# Fault Detection and Diagnosis with Imbalanced and Noisy Data: A Hybrid Framework for Rotating Machinery

**Masoud Jalayer** [1,2,*] , **Amin Kaboli** [3], **Carlotta Orsenigo** [2] and **Carlo Vercellis** [2]

1 Department of Mechanical Engineering, University of Victoria, Victoria, BC V8P 5C2, Canada

2 Department of Management, Economics and Industrial Engineering, Politecnico di Milano, Via Lambruschini 4/b, 20156 Milan, Italy; carlotta.orsenigo@polimi.it (C.O.); carlo.vercellis@polimi.it (C.V.)

3 Institute of Mechanical Engineering, School of Engineering, Swiss Federal Institute of Technology at Lausanne (EPFL), 1015 Lausanne, Switzerland; amin.kaboli@epfl.ch

* Correspondence: masoudjalayer@uvic.ca

**Abstract:** Fault diagnosis plays an essential role in reducing the maintenance costs of rotating machinery manufacturing systems. In many real applications of fault detection and diagnosis, data tend to be imbalanced, meaning that the number of samples for some fault classes is much less than the normal data samples. At the same time, in an industrial condition, accelerometers encounter high levels of disruptive signals and the collected samples turn out to be heavily noisy. As a consequence, many traditional Fault Detection and Diagnosis (FDD) frameworks get poor classification performances when dealing with real-world circumstances. Three main solutions have been proposed in the literature to cope with this problem: (1) the implementation of generative algorithms to increase the amount of under-represented input samples, (2) the employment of a classifier being powerful to learn from imbalanced and noisy data, (3) the development of an efficient data preprocessing including feature extraction and data augmentation. This paper proposes a hybrid framework which uses the three aforementioned components to achieve an effective signal based FDD system for imbalanced conditions. Specifically, it first extracts the fault features, using Fourier and wavelet transforms to make full use of the signals. Then, it employs Wasserstein Generative Adversarial with Gradient Penalty Networks (WGAN-GP) to generate synthetic samples to populate the rare fault class and enrich the training set. Moreover, to achieve a higher performance a novel combination of Convolutional Long Short-term Memory (CLSTM) and Weighted Extreme Learning Machine (WELM) is also proposed. To verify the effectiveness of the developed framework, different bearing datasets settings on different imbalance severities and noise degrees were used. The comparative results demonstrate that in different scenarios GAN-CLSTM-ELM significantly outperforms the other state-of-the-art FDD frameworks.

**Keywords:** fault detection; rotating machinery; condition monitoring; generative adversarial networks; signal processing; predictive maintenance

## 1. Introduction

Rotating machinery is one of the essential equipment in today's industrial environments. From petroleum, automobile, chemicals, pharmaceutical, mining, and power generation plants to consumer goods, at least there is a machine with a rotating component. The rotating component could be the gearbox, axles, wind, steam and gas turbines, centrifugal and oil-free screw compressors, and pumps. A total of 30% of rotating machinery breakdowns are mainly caused by loose, partially rubbed, misaligned, cracked, and unbalanced rotating parts [1]. Machine breakdowns can present complex challenges during day-to-day operations and significantly impact business profitability and operations productivity. Monitoring machine health conditions can prevent machine breakdowns and reduce the maintenance costs of manufacturing systems [2]. It is, hence, crucial to

develop efficient diagnosis systems to analyze different health conditions of the rotating components.

There are two main approaches for coping with fault detection and diagnosis in rotating machinery: (1) physical-based control systems and (2) data-driven-based models. Recent advancements in computer processing and digital technologies enhanced the robustness and higher computational capabilities to use data-driven fault detection and diagnosis models. Implementing these models enable us to monitor and control the parameters of machines from a remote distance and drive insights. That is the main reason for which data-driven fault detection and diagnosis models are used in smart manufacturing systems [2]. Figure 1 shows some expected steps that a practitioner should take to practice a data-driven fault detection and diagnosis.

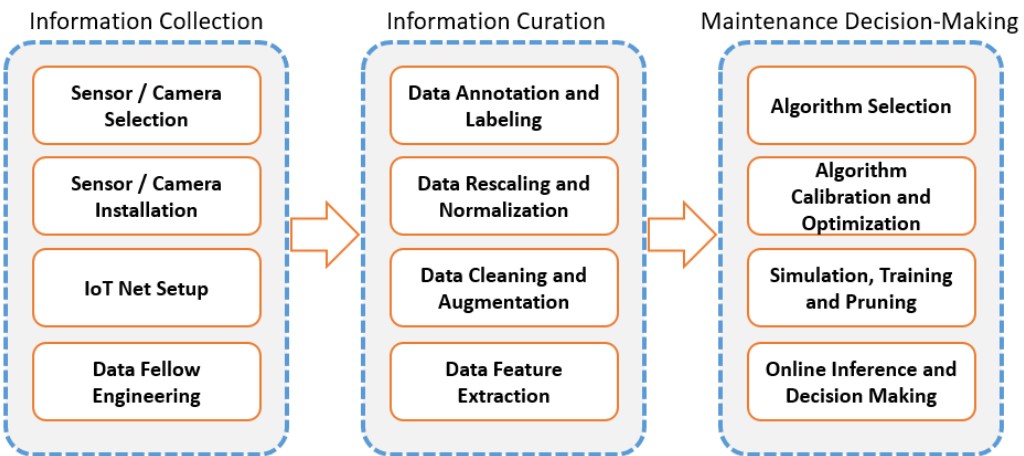

**Figure 1.** Main steps of an automated Fault Detection and Diagnosis system.

The main contributions of this paper are as follows: (1) In order to get higher classification performance in different environments, a hybrid deep learning architecture is designed such that it takes Fourier and Wavelet spectra of the vibration signals. This architecture uses CNN blocks to find shift-agnostic characteristics of the fault types, a LSTM block which understands the spatiotemporal and sequential features of it and, finally, a Weighted ELM classifier which is effective in learning from scarce patterns, the necessity of which is examined through experimental comparisons. The proposed classifier is named CLSTM-ELM. (2) A Wasserstein–GAN model with a gradient penalty is developed and employed in the hybrid framework to reproduce rare patterns and enhance the training set. The effectiveness of this proposition is investigated in Section 5. (3) A comprehensive set of scenarios is designed to study the effect of different imbalance severities and noise degrees on the performance of the framework. A sensitivity analysis is conducted on the scenarios revealing more insights about the characteristics of the model. (4) Seven state-of-the-art FDD models are chosen to compete with the proposed framework on four different dataset settings. The experimental comparison illustrates how implementing WGAN-GP and W-ELM can improve the classifier performance and shows the superiority of GAN-CLSTM-ELM over other algorithms.

The rest of the paper is organized as follows. Section 2 provides an overview of the principal AI-based approaches proposed for FDD problems. In Section 3, the theory behind WGAN-GP, LSTM, Convolutional layers and W-ELM is briefly reviewed. Then, the proposed hybrid framework, GAN-CLSTM-ELM, is presented in Section 4. Section 5 compares the performance of different FDD algorithms on different imbalance ratios and noise severities. Finally, some research conclusions and future extensions are provided in Section 6.

## 2. Review of Current Models

Early data-driven fault detection and diagnosis (hereafter FDD) models have benefited from traditional Artificial Intelligence (AI) models, or "shallow learning" models, such as Support Vector Machines (SVM), Decision Trees (DT), and Multi-layer Perceptron (MLP) [3]. Despite the applicability of traditional AI models to FDD problems, these models show poor performances and limitations when dealing with complicated fault patterns such as the above-mentioned rotating machinery faults [4]. One of the first applications of rotating machinery FDD dates back to 1969 in Boeing Co., when Balderston [5] illustrated some characteristics of the fault signs on the signals measured by an accelerometer in natural and high frequencies. Ref. [6] employed the rectified envelope signals with a synchronous averaging, which was later called "envelope analysis", to identify bearing local faults. The peak localization in the vibration signal spectrum is another classical example of the fault detection methods for the ball bearing faults [7].

Recently, with the emergence of novel deep learning architectures and their promising pattern recognition capabilities, many researchers proposed deep learning solutions for data-driven-based FDD systems [8]. These FDD approaches rely on the common assumption that the distribution of classes for different machine health conditions is approximately balanced. In practice, however, the number of instances may significantly differ from a fault class to another. This causes a crucial issue since a classifier which has been trained on such a data distribution primarily exhibits a skewed accuracy towards the majority class, or fails to learn the rare patterns. Most of the proposed FDD approaches, thus, suffer from higher misclassification ratios when dealing with scarce conditions such as in high-precision industries where the number of faults are limited [9].

Through their deep architectures, deep learning-based methods are capable of adaptively capturing the information from sensory signals through non-linear transformations and approximate complex non-linear functions with small errors [3]. Auto-encoders (AE) are among the most promising deep learning techniques for automatic feature extraction of mechanical signals. They have been adopted in a variety of FDD problems in the semiconductor industry [10], foundry processes [11], gearboxes [12] and rotating machinery [13,14]. Ref. [15] employed the "stacked" variation of AE to initialize the weights and offsets of a multi-layer neural network and to provide an expert knowledge for spacecraft conditions. However, to cope with mechanical signals, using a single AE architecture has shown some drawbacks: it may only learn similar features in feature extraction and the learned features may have shift variant properties which potentially lead to misclassification. Some approaches were proposed to make this architecture appropriate for signal-based fault diagnosis tasks. Ref. [16] used a local connection network on a normalized sparse AE, called NSAE-LCN, to overcome these shortcomings. Ref. [17] developed a stacked-AE to directly learn features of mechanical vibration signals on a motor bearing dataset and a locomotive bearing dataset; specifically, they first used a two-layer AE for sparse filtering and then applied a softmax regression to classify the motor condition. The combination of these two techniques let the method achieved high accuracy in bearing fault diagnosis.

Extreme learning machine (ELM) is a competitive machine learning technique, which is simple in theory and fast in implementation. As an effective and efficient machine learning technique, ELM has attracted tremendous attention from various fields in recent years. Some researchers suggest ELM and Online Sequential ELM (OS-ELM) for learning from imbalance data [18–20]. ELM and OS-ELM can learn extremely fast due to their ability to learn data one-by-one or chunk-by-chunk [21]. Despite their effective performances on online sequential data, the performance associated to their classical implementation on highly imbalanced data is controversial; according to [22], for example, OS-ELM tends to have poor accuracy on such data. Therefore, they proposed a voting-based weighted version of it, called VWOS-ELM, to cope with severely rare patterns, whereas [9] developed a two-stage hybrid strategy using a modified version of OS-ELM, named PL-OSELM. In offline stage, the principal curve is employed to explore the data distribution and develop an initial model on it. In online stage, some virtual samples are generated according to

the principal curve. The algorithm chooses virtual minority class samples to feed more valuable training samples.

Considering the promising results obtained by ELM-based classifiers coping with imbalanced data, they accordingly became one of the mainstreams in FDD research area. In [23], the authors developed an evolutionary OS-ELM for FDD for bearing elements of high-speed electric multiple units. They employed a K-means synthetic minority oversampling technique (SMOTE) for oversampling the minority class samples. They also used an artificial bee colony (ABC) algorithm to find a near-optimum combination of input weights, hidden layer bias, and number of hidden layer nodes of the OS-ELM. In another paper, Ref. [24] used density-weighted one-class ELM for fault diagnosis in high-voltage circuit breakers (HVCBs), using vibration signals. Ref. [25] applied an adaptive class-specific cost regulation ELM (ACCR-ELM) with variable-length brainstorm algorithm for its parameter optimization to conveyor belt FDD. The proposed algorithm exhibits a stable performance under different imbalance ratios. Ref. [26] presented a feature extraction scheme on time-domain, frequency-domain, and time-frequency-domain, to feed a full spectrum of information gained from the vibration signals to the classifier. They also demonstrated that the cost-sensitive gradient boosting decision tree (CS-GBDT) shows a satisfactory performance for imbalanced fault diagnosis. In another FDD framework for rolling bearings [27], the authors coupled an Optimized Unsupervised Extreme Learning Machine (OUSELM) with an Adaptive Sparse Contractive Auto-encoder (ASCAE). The ASCAE can gain an effectual sparse and more sensitive feature extraction from the bearing vibration signals. A Cuckoo search algorithm was also proposed to optimize the ELM hyper-parameters. Another variation of ELM was developed by [28] to deal with imbalanced aircraft engines fault data which are derived from the engine's thermodynamic maps. This ELM variation flexibly sets a soft target margin for each training sample; hence, it does not need to force the margins of all the training samples exactly equaling one from the perspective of margin learning theory. After some experiments on different datasets, including the aircraft engine, it is concluded that SELM outperforms ELM.

On the other hand, there are frameworks for imbalanced and noisy FDD without the employment of any ELM variation. Ref. [16] proposed a Deep Normalized Convolutional Neural Network (DNCNN) for FDD under imbalanced conditions. The DNCNN employs a weighted softmax loss which assumes that the misclassification errors of different health conditions share an equivalent importance. Subsequently, it minimizes the overall classification errors during the training processes and achieves a better performance when dealing with imbalanced fault classification of machinery adaptively. Ref. [29] used WGAN-GP to interpolate stochastically between the actual and virtual instances so that it ensures that the transition region between them is stable. They also utilized a Stacked-AE to classify the enhanced dataset and determined the availability of the virtual instances. Since a single GAN model encounters hardship and poor performance when dealing with FDD datasets, Ref. [30] proposed a framework based on GANs under small sample size conditions which boost the adaptability of feature extraction and consequently diagnosis accuracy. The effectiveness and satisfactory performance of the proposed method were demonstrated using CWRU bearing and gearbox datasets. Another novel GANs-based framework, named dual discriminator conditional GANs (D2CGANs), has been recently proposed to learn from the signals on multi-modal fault samples [31]. This framework automatically synthesizes realistic high-quality fake signals for each fault class and is used for data augmentation such that it solves the imbalanced dataset problem. After some experiments on the CWRU bearing dataset, the authors showed that Conditional-GANs, Auxiliary Classifier-GANs and D2CGANs significantly outperform GANs and Dual-GANs. Ref. [32] proposed a framework which adopts a CNN-based GANs with the coordinative usage of two auxiliary classifiers. The experimental results on analog-circuit fault diagnosis data suggested that the proposed framework achieves a better classification performance than that of DBN, SVM and artificial neural networks (ANN). Ref. [33] presented a CNN-based GANs for rotating machinery FDD which uses a Wavelet Transform (WT) technique.

The proposed so-called WT-GAN-CNN approach extracts time-frequency image features from one-dimension raw signals using WT. Secondly, GANs are used to generate more training image samples while the built CNN model is used to accomplish the FDD on the augmented dataset. The experimental results demonstrated high testing accuracy in the interference of severe environment noise or when working conditions were changed.

### 3. Background Theory
*3.1. WGAN-GP*

In the classical definition, GANs consist of two adversarial networks trained in opposition to one another: (1) a generative model, $G$, which learns the data distribution to generate a fake sample $\tilde{x}^{(i)} = G(z)$ from a random vector $z$, where $z \sim \mathscr{P}_z$ and $\mathscr{P}_z$ is the noise distribution; (2) a discriminator model, $D$, which determines if the sample is generated by $G$ or is a real sample. $G$ strives to deceive $D$ by making realistic random samples, while $D$ receives both real and fake samples. On the contrary, $D$ tries to find out the source of each sample by calculating its corresponding probability, $p(S|x) = D(x)$, and is trained to maximize the log-likelihood it assigns to the correct source [34]

$$\min_{G} \max_{D} V(D, G) = \mathbb{E}_{x \sim \mathscr{P}_r}[\log(D(x))] + \mathbb{E}_{\tilde{x} \sim \mathscr{P}_f}[\log(1 - D(\tilde{x}))], \tag{1}$$

where $\mathscr{P}_r$ and $\mathscr{P}_f$ denote the distribution of the raw data and of the fake samples, respectively. Then, the model reaches a dynamic equilibrium if $\mathscr{P}_f = \mathscr{P}_r$ [34].

While GAN is a powerful generative model, it suffers from training instability. Some different solutions have been proposed to solve this problem. Wasserstein GAN (WGAN) is one of the novel proposed techniques offering a new loss function which has demonstrated a better performance and a better model stability. The present paper uses a variation of GAN, Entropy-based WGAN-GP proposed by [35], generating an entropy-weighted label vector for each class with respect to its frequency.

When the discriminator $D$ is sufficiently trained, the gradient of the generator $G$ is relatively small; when the effect of $D$ is lower, it gets larger. WGAN employs a distance called Wasserstein to calculate the difference between the distributions of the real and fake samples [36], which can be mathematically written as follows

$$\mathcal{W}(\mathscr{P}_r, \mathscr{P}_f) = \inf_{\lambda \in \Pi(\mathscr{P}_r, \mathscr{P}_f)} \mathbb{E}_{(a,b) \sim \lambda}[\|a - b\|], \tag{2}$$

where $\Pi(\mathscr{P}_r, \mathscr{P}_f)$ denotes the set of all distributions with margins of $\mathscr{P}_r$ and $\mathscr{P}_f$, and $\lambda(a, b)$ represents the distance between two given distributions, $a$ and $b$. Therefore, the Wasserstein variable can be interpreted as the transportation cost between the distributions of real and fake datasets. To avoid the gradient uninformativeness issue and to guarantee the existence and uniqueness of the optimal discriminative function and the respective Nash equilibrium, Lipschitz condition is applied [37]. In the proposed WGAN, the discriminative function is restricted to 1-Lipschitz. The WGAN, hence, proposes a revised objective function based on (1), using Kantorovich-Rubinstein duality, and is formulated as

$$\min_{G} \max_{D \in \mathcal{L}_1} \mathbb{E}_{x \sim \mathscr{P}_f}[D(x)] - \mathbb{E}_{\tilde{x} \sim \mathscr{P}_r}[D(\tilde{x})], \tag{3}$$

where $\mathcal{L}_1$ is the collection of 1-Lipschitz functions. In this case, under an optimal discriminator which minimizes the objective function with respect to the parameters of $G$, the model strives to minimize $\mathcal{W}(\mathscr{P}_r, \mathscr{P}_f)$.

Let $\delta \sim U[0, 1]$ be a random number to have a linear interpolation between $x$ and $\tilde{x}$:

$$\hat{x} = \delta x + (1 - \delta)\tilde{x}, \tag{4}$$

Therefore, the loss function of the discriminator, $\mathcal{L}_D$, can be determined as follows

$$\mathcal{L}_D = \mathbb{E}_{\tilde{x} \sim \mathscr{P}_g}[D(\tilde{x})] - \mathbb{E}_{x \sim \mathscr{P}_r}[D(x)] + \gamma \mathbb{E}_{\hat{x} \sim \mathscr{P}_z}[(\|\nabla_{\hat{x}} D(\hat{x})\|_2 - 1)^2], \tag{5}$$

where $\gamma$ stands for the gradient penalty coefficient. The last part of Equation (5) denotes the gradient penalty: $\gamma \mathbb{E}_{\hat{x} \sim \mathscr{P}_z}[(\|\nabla_{\hat{x}} D(\hat{x})\|_2 - 1)^2]$ [38].

Figure 2 exhibits the simplified process of synthesizing rotating machinery signal samples out of some random noises in a conventional GAN model.

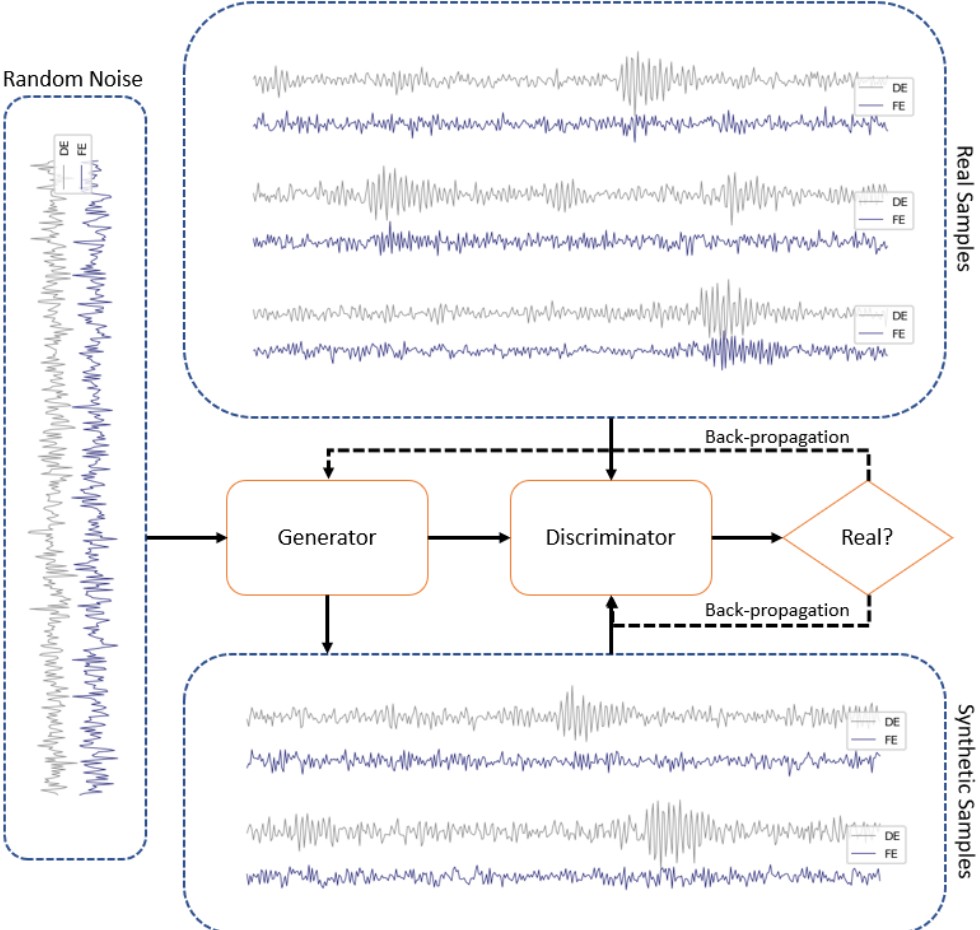

**Figure 2.** The schematic process in GANs.

### 3.2. CLSTM

RNN (Recurrent Neural Network) is a powerful class of artificial deep learning architectures proposed to identify patterns in sequential data. It can consider time and sequence by taking the present data and the recent past data as inputs. RNN is trained across the time steps using backpropagation. However, due to the multiplication of gradients at time steps the gradient value may vanish or blow up rapidly. This issue limits its usage when the time window is greater than 10 discrete time steps [39]. By adding constant error carousels and introducing forget gates to RNN, a new form of RNN architecture is proposed, named LSTM. These adopted forget gates are able to control the utilization of information in the cell states and impede the vanishing or exploding gradient issues [40]. Compared to RNN, LSTM is more powerful in capturing long-term dependencies of the data features where it can handle time-windows exceeding 1000 discrete time stamps [39].

On the other hand, convolutional neural networks are mainly composed of convolutional, pooling and normalization layers, making them capable of understanding the data shift-invariance and sharing the weights through convolutional connections. This weight

sharing makes CNN lighter for computation since it reduces the number of parameters in the network.

Let $x_i = [\kappa_1, \ldots, \kappa_L]$ be the sequential data, where $L$ denotes the length of the signal sample, and $\kappa_i \in \mathbb{R}^d$ the set of values at each timestamp, where $d$ is the number of channels. The convolution operation is defined as the following dot product:

$$c_i = \varphi(u \cdot \kappa_{i:i+m-1} + b), \tag{6}$$

where $u \in \mathbb{R}^{md}$ is a filter vector, $b$ is the bias, $\varphi$ is an activation function, and $\kappa_{i:i+m-1}$ represents an $m$-length window starting from the $i$th time stamp of the sample. Sliding the filter window from the first timestamp of the sample to its last possible one, a feature map is given as follows:

$$\mathscr{C}_\xi = [c_1, c_2, \ldots, c_{L-m+1}], \tag{7}$$

where $\mathscr{C}_\xi$ corresponds to the $\xi$th filter.

To reduce the length of these feature maps and minimizing the model parameters, pooling layers are proposed. Max Pooling layer is one of the most common pooling techniques. The compressed feature vector, $\mathscr{H}$, is defined as follows:

$$\mathscr{H} = [h_1, h_2, \ldots, h_{\left(\frac{L-m}{s}\right)+1}], \tag{8}$$

where $h_\xi = \max(\mathscr{C}_{(\xi-1)s}, \mathscr{C}_{(\xi-1)s+1}, \ldots, \mathscr{C}_{(\xi s-1)})$ on the $s$ consecutive values of feature map $\mathscr{C}_\xi$ and $s$ denotes the pooling length. Batch normalization is widely used in CNN blocks to reduce the shift of interval covariance and make the learning quicker by alleviating the computational load. The normalization is completed by making each individual scalar feature with zero mean and unit variance. This process can be mathematically described as:

$$\hat{\kappa}_i = (\kappa_i - \mathbb{E}[x_i])\sqrt{Var(x_i) + \epsilon}, \tag{9}$$

where $\epsilon$ is a small constant added for numerical stability. However, the extracted features can be affected when the features of a certain layer are normalized directly by Equation (9), leading to poor network expression abilities. To resolve this issue, each normalized value $\kappa_i$ is modified based on the scale parameter $\varrho_i$ and the shift parameter $\varpi_i$. These two learnable reconstruction parameters can recover the feature distribution of the original network. The following formula can be used to determine the output of the neuron response:

$$\hat{v}_i = \varrho_i \hat{\kappa}_i + \varpi_i. \tag{10}$$

*3.3. W-ELM*

ELM is in general a single-hidden-layer feed-forward neural network (SLFN). The difference between SLFN and ELM lies in how the weights of hidden layer and output layer neurons are updated. In SLFN, the weights of both input and outputs layers are initialized randomly, and the weights of both the layers are updated by the backpropagation algorithm. In ELM, the weights of the hidden layers are assigned randomly but never updated, and only the weights of the output layer are updated during the training process. As in ELM, the weights of only one layer are to be updated as opposed to both layers of SLFN; this makes ELM faster than SLFN.

Let the training dataset be $\{(x_i, y_i)\}_{(i=1)}^N$, where $x_i$ is the input vector and $y_i$ is the output vector. The output of the $j$th hidden layer neuron is given by $\varphi_a(a_j, b_j, x_i)$, where $a_j$ is the weight vector connecting the input neurons to the $j$th hidden layer neuron, $b_j$ is the bias of the $j$th hidden neuron, and $\varphi_a$ is the activation function. Each hidden layer neuron of ELM is also connected to each output layer neuron with some associated weights. Let

$\beta = [\beta_1, \beta_2, \ldots, \beta_K]^T$ denote the output weights connecting the hidden layer (composed of $K$ hidden nodes) with output neurons. Thus, the $i$th output is determined as:

$$o_i = \sum_{j=1}^{K} \beta_j \phi_a(a_j, b_j, x_i), \quad i = 1, \ldots, N. \tag{11}$$

Let $H = (H_{ij}) = (\phi_a(w_j, b_j, x_i))$ be the hidden layer matrix. The $N$ equations of the output layer (Equation (11)) can be shortly written as follows:

$$O = H\beta. \tag{12}$$

Using Moore–Penrose generalized inverse [41], $H^\dagger$, a least square solution, referred to as the extreme learning machine, can be determined mathematically as follows:

$$\beta = H^\dagger Y = \begin{cases} H^T(\frac{1}{C} + HH^T)^{-1}Y & N < K \\ (\frac{1}{C} + H^TH)^{-1}H^TY & N \geq K, \end{cases} \tag{13}$$

where $C$ is a positive parameter to achieve a better generalization performance [42]. Weighted ELM [20] considers a $N \times N$ diagonal matrix $W$ associated with each training sample $x_i$. If $x_i$ belongs to the minority class, the algorithm allocates a relatively larger weight to $w_i$ rather than those of majority classes, which intensifies the impact of minority classes in the training phase. Therefore, the solution of Equation (12) will be obtained by using the optimization formula of ELM:

$$Minimize : L_{pELM} = \frac{1}{2}\|\beta\|^2 + CW\frac{1}{2}\sum_{i=1}^{N}\|\eta_i\|^2 \tag{14}$$

$$subject\ to : h(x_i)\beta = y_i^T - \eta_i^T, i = 1, \ldots, N.$$

According to the KKT theorem [43], the solution to Equation (14) is obtained as follows:

$$\beta = H^\dagger Y = \begin{cases} H^T(\frac{1}{C} + WHH^T)^{-1}WY & N < K \\ (\frac{1}{C} + H^TWH)^{-1}WH^TY & N \geq K. \end{cases} \tag{15}$$

## 4. The Proposed FDD Model

As mentioned in Section 1, there is a need to improve the performance of imbalanced and noisy FDD systems. Therefore, this paper presents a hybrid framework which embeds three steps: (1) the employment of a generative algorithm to improve the training set, (2) the signal processing with FFT and CWT techniques providing the deep learning classifier a deeper understanding of the fault's identity, (3) the development of a hybrid classifier based on CNN, LSTM and weighted ELM, as illustrated in Figure 3.

### 4.1. Sample Generation Model Design

The structure of the WGAN-GP generator $G$ comprises a five-layered autoencoder of $l, \frac{l}{2}, \frac{l}{4}, \frac{l}{2}$ and $l$ neurons, while the discriminator $D$ is composed of three convolutional-LSTM blocks. The input variable $z$ has a dimension of $l \times 1$. Due to the poor performance in weight clipping of WGAN-GP, the paper uses an alternative in the form of a gradient penalty in the discriminator loss function, which has been introduced by [44] and which achieves high performances compared to other GAN models. Algorithm 1 shows how WGAN-GP works, where $\rho$ is the gradient coefficient, $\chi_d$ is the number of discriminator iterations per each generator iteration, $\chi_b$ denotes the batch size, $\vartheta, \mu_1$ and $\mu_2$ are the Adam hyper-parameters, and $\omega_1$ and $\theta_1$ represent the initial discriminator and generator parameters, respectively.

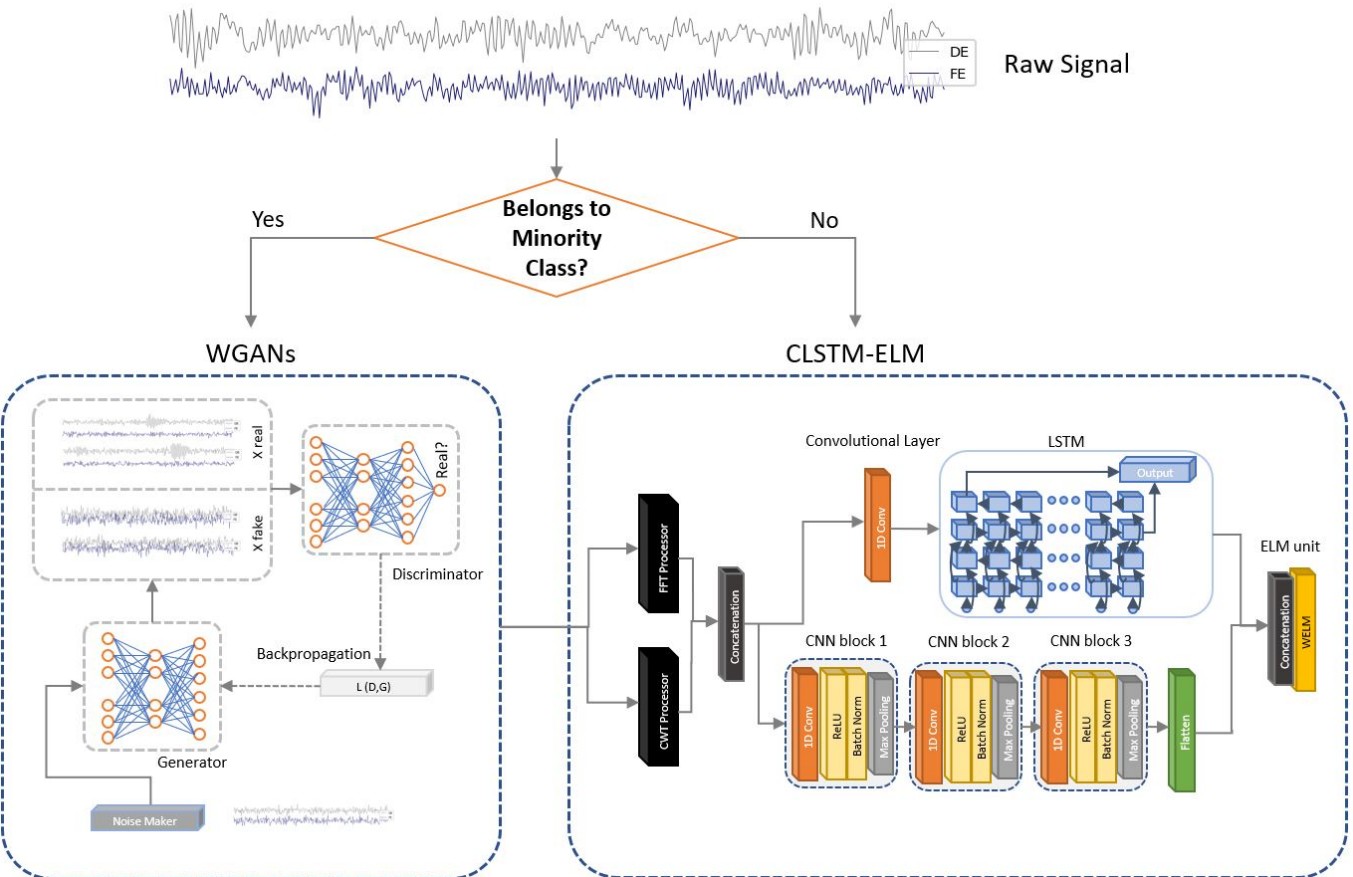

**Figure 3.** Schematic illustration of the proposed model for the training set.

---

**Algorithm 1:** WGAN-GP

**Input:** $\gamma, n_{critic}, m, \alpha, \beta_1, \beta_2, \omega_0, \theta_0$

1 **while** $\theta$ *is not converged* **do**

2    **for** $t \leftarrow 1, ..., n_{critic}$ **do**

3      **for** $i \leftarrow 1, ..., m$ **do**

4        Sample from real dataset $x \sim \mathscr{P}_r$,

5        Generate noise samples $z \sim \mathscr{P}_z$,

6        Generate a random number $\delta \sim U[0,1]$

7        $\tilde{x} \leftarrow G_\theta(z)$

8        $\hat{x} \leftarrow \delta x + (1 - \delta)\tilde{x}$

9        $\mathscr{L}^{(i)} \leftarrow D_\omega(\tilde{x}) - D_\omega(x) + \gamma(\|\nabla_{\tilde{x}} D_\omega(\hat{x})\|_2 - 1)^2$

10      $\omega \leftarrow Adam(\nabla_\omega \frac{1}{m} \sum_{i=1}^{m} L^i, \omega, \alpha, \beta_1, \beta_2)$

11    Sample batch of $m$ noise samples $\{z^i\}_{i=1}^{m} \sim \mathscr{P}_z$

12    $\theta \leftarrow Adam(\nabla_\theta \frac{1}{m} \sum_{i=1}^{m} -D_\omega(G_\theta(z)), \omega, \alpha, \beta_1, \beta_2)$

---

### 4.2. Fault Diagnosis Model Design

In order to reveal more information about the fault characteristics, the paper separately employs two signal processing feature extraction techniques on the input samples: FFT, based on one-dimensional $n$-point discrete Fourier Transform (using Scipy FFT library (https://docs.scipy.org/doc/scipy/reference/fft.html (accessed on 22 March 2022))), and CWT [45], based on the Mexican hat wavelet transform with a width of three (using Scipy CWT library (https://docs.scipy.org/doc/scipy/reference/generated/scipy.signal.cwt.html (accessed on 22 March 2022))) on the short bursts that are collected from raw sensory signals. Figure 4 illustrates the extracted features on some random raw samples

from a bearing dataset. The transform outputs are then merged to be passed to the deep learning architectures. As it is demonstrated in [46], employing these two feature extraction techniques significantly improves the diagnosis performance and increases the accuracy.

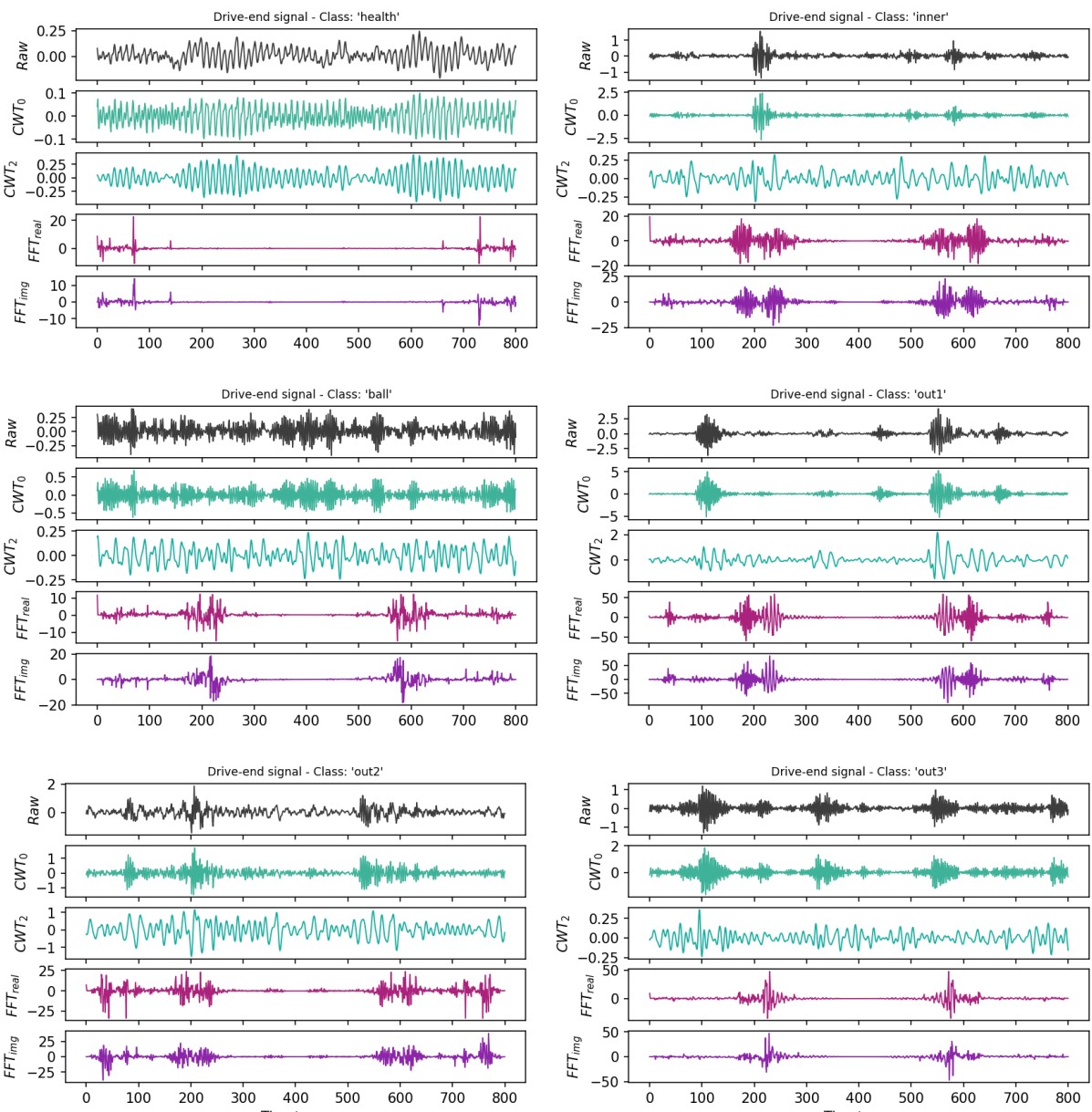

**Figure 4.** Illustration of the extracted features from some raw signals.

As it is shown in Figure 3, the paper proposes a dual-path deep learning architecture which combines LSTM and CNN. The reason behind this duality lies in the nature of these two architectures and the fact that each of them explains a different feature type. More specifically, Ref. [47] illustrated that the concatenation of CNN and LSTM features meaningfully enhances the classification accuracy.

Table 1 shows the specifications and parameters for each layer in the proposed diagnosis architecture. In the first pathway, after applying a one-dimensional convolutional layer on the pre-processed input tensors which extracts the local and discriminative features, an LSTM is added to encode the long-term temporal patterns. The importance of adding a convolutional layer prior to the LSTM layer is not only that it reduces the high-frequency noise impact, but it also helps the LSTM to learn more effectively. The convolutional layer

processes the sequence of the features extracted after FFT and CWT. The model slides the kernel filters on the sequence and generates feature maps. These feature maps are subsequently processed by an LSTM which acquires the spatial dependencies of these sequenced features.

**Table 1.** Deep learning layer specifications of the diagnosis architecture.

| Block | Layer | Specifications |
| --- | --- | --- |
| CNN Block1 | 1D-Convolutinoal<br>Batch Normalization<br>1D-Max Pooling | filters: 16; kernel size: 5; stride: 2; padding: 0<br>momentum: 0.99; epsilon: 0.001<br>pool size: 2, padding: 0, stride: 1 |
| CNN Block2 | 1D-Convolutinoal<br>Batch Normalization<br>1D-Max Pooling | filters: 32; kernel size: 3; stride: 1; padding: 0<br>momentum: 0.99; epsilon: 0.001<br>pool size: 2, padding: 0, stride: 1 |
| CNN Block3 | 1D-Convolutinoal<br>Batch Normalization<br>1D-Max Pooling | filters: 64; kernel size: 3; stride: 1; padding: 0<br>momentum: 0.99; epsilon: 0.001<br>pool size: 2, padding: 0, stride: 1 |
| Convolutional LSTM | 1D-Convolutinoal<br>Embedding<br>LSTM | filters: 20; kernel size: 8; stride: 3; padding: 0<br>input dimension: 200; output dimension: 64<br>units: 64; activation function: tanh; |
| ELM unit | WELM | #nuerons: 150; activation function: sigmoid; |

On the other hand, in the second pathway three one-dimensional CNN blocks are adjoined to better extract the local features of the Fourier and Wavelet transform-based diagrams. Each CNN block contains a convolutional layer which convolves the local regions of the diagrams, and a Rectified Linear Unit (ReLU) activation function, which helps the network achieve a non-linear expression and, consequently, make the learned features more distinguishable. To reduce the computational complexity and decrease the covariance of shift intervals, a batch normalization layer is added to each CNN block, following by a max pooling layer which compresses the learnt features, advances its local translation invariance and also alleviates the learning computational expenses. These CNN blocks are followed by a flatten layer that reshapes the tensor size to a vector to become compatible to join the output of the first pathway and to be fed to the classifier.

For the classification architecture, the paper employs a weighted ELM which is introduced in Section 3.3. ELM classifiers can get fast training speeds by means of non-tuned training strategy. They also tend to have high generalization performance in multi-class problems and showed excellent classification performance in different studies [19]. Compared to unweighted ELM, weighted ELM is aimed to put an additional accent on the samples implying the imbalanced class distribution, so that the features in samples from the minority class are also well perceived by the ELM [20]. Therefore, after concatenating the outputs of both pathways, their combined learnt features are passed to the W-ELM to diagnose the fault types.

### 4.3. General Procedure of the Proposed Model

The schematic flowchart of the proposed intelligent FDD model is illustrated in Figure 3. The general procedure of this model is summarized as follows:

- **Step 1**: The sensory signals are collected from the accelerometers mounted on the rotating machinery.
- **Step 2**: The training, the test, and the validation datasets are constructed from the raw signals to separate bursts by resampling.
- **Step 3**: The training dataset is augmented using WGAN-GP introduced in Section 3.1 on the minority classes. The fake samples are added to the real samples to make the training dataset balanced.

- **Step 4**: By employing FFT and CWT techniques the model can extract fault signatures which were hidden in the raw signals. The extracted Fourier and Wavelet transform-based diagrams are concatenated to form three-dimensional tensors (such as Figure 5) which will be given in input to the deep learning blocks.
- **Step 5**: These pre-processed samples go through two different paths of deep learning blocks: (1) a one-dimensional convolutional layer followed by an LSTM block, and (2) three blocks of CNN architectures followed by flatten and dense layers.
- **Step 6**: After concatenating the outputs of the two deep learning paths, a W-ELM technique is used to classify the extracted deep features and diagnose the fault type.

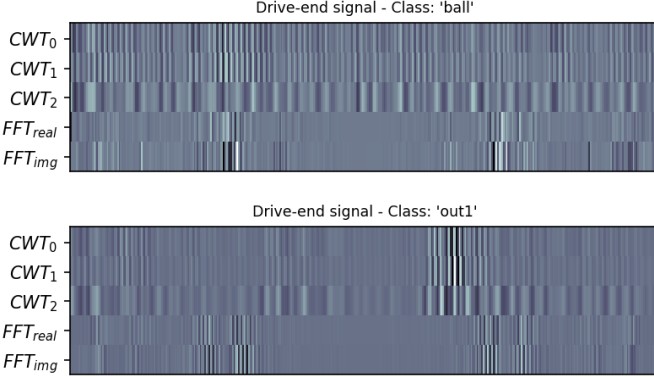

**Figure 5.** The output of Step 4—Two tensors concatenating the extracted features.

## 5. Results

To evaluate the effectiveness of the proposed method, some experiments were run on one of the most widely used bearing fault datasets, known as Case Western Reserve University (CWRU) bearing dataset (https://csegroups.case.edu/bearingdatacenter/home (accessed on 22 March 2022)). To conduct a comprehensive comparison, we defined different noise and imbalance conditions on which eight different deep learning-based FDD methods were tested. All experiments were performed by using Python 3.9 on a computer with a GPU of NVIDIA Geforce GTX 1070 with CUDA version of 10.1 and 16 GB of memory.

### 5.1. Dataset Description

The paper employs CWRU bearing dataset using the test stand shown in Figure 6, that consists of a motor, a torque transducer/encoder, a dynamometer, and control electronics. The dataset consists of five different types of faults corresponding to inner race, balls and outer race in three different orientations: 3 o'clock (directly in the load zone), 6 o'clock (orthogonal to the load zone) and 12 o'clock (opposite to the load zone). Moreover, the faults are collected in a range of severity varying between 0.007 inches to 0.040 inches in diameter. The dataset is also recorded for motor loads, from 0 to 3 horsepower. However, for the sake of simplicity this paper uses only one motor speed of 1797 RPM. The samples are collected at 12,000 samples/second frequency from two accelerometers mounted on fan-end and drive-end of the machine. In the experiments we took signal bursts of 800 timestamps, equal to 66.6 milli-seconds, to generate some different datasets of approximately 25,500 signal bursts.

To explore the diagnostic capabilities of the proposed framework in imbalanced conditions, some non-equitant sets of samples were selected such that a fault class becomes rare. Table 2 shows the distribution of samples for each machine condition in the selected sets, where the value of $\alpha$ denotes the percentage of minority class within the whole dataset. Accordingly, as $\alpha$ decreases the imbalance degree increases. In this paper we chose "out3" class to represent the minority class, whose samples correspond to the outer race faults of opposite load zone position. In these scenarios, the "health" class, corresponding to the healthy condition, represents $(80 - \alpha)$ percent of the whole dataset, while the other fault classes account for 5% each. The generative algorithm, subsequently, strives to equalize the sample size of the fault classes in the training set by augmenting the minority class.

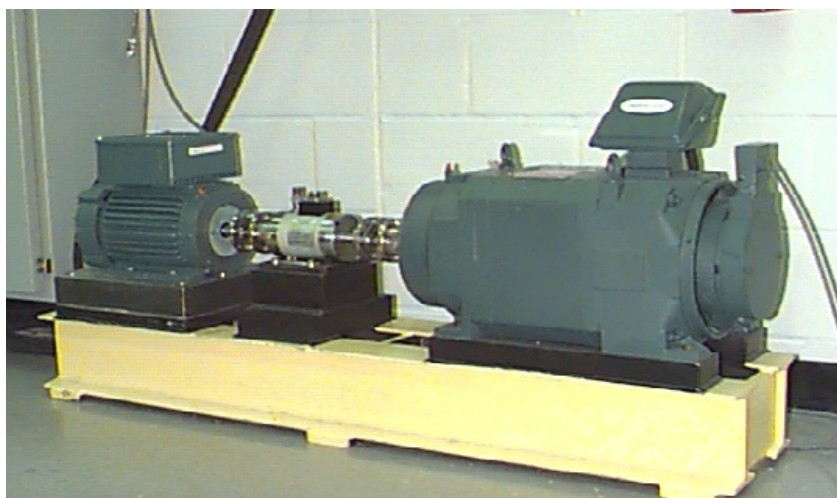

**Figure 6.** Two-horsepower (**left**), a torque transducer and encoder (**center**) and a dynamometer (**right**) used to collect the dataset.

**Table 2.** The distribution of condition type samples in the cases.

| Minority Share (%) | Percentage of Training Samples in Each Condition | | | | | |
|---|---|---|---|---|---|---|
| | **Health** | **Inner** | **Ball** | **Out1** | **Out2** | **Out3** |
| $\alpha = 4$ | 76% | 5% | 5% | 5% | 5% | 4% |
| $\alpha = 2$ | 78% | 5% | 5% | 5% | 5% | 2% |
| $\alpha = 1$ | 79% | 5% | 5% | 5% | 5% | 1% |
| $\alpha = 0.5$ | 79.5% | 5% | 5% | 5% | 5% | 0.5% |
| $\alpha = 0.25$ | 79.75% | 5% | 5% | 5% | 5% | 0.25% |

Adding "additive white Gaussian noise" with different signal-to-noise ratios (SNRs) to the original samples, the paper is able to examine the performance of GAN-CLSTM-ELM framework on different natural noise severity levels. These noisy samples better portray the real-world industrial production settings where the noise varies a lot. The original drive-end and fan-end signals with their driven noisy samples are exhibited in Figure 7.

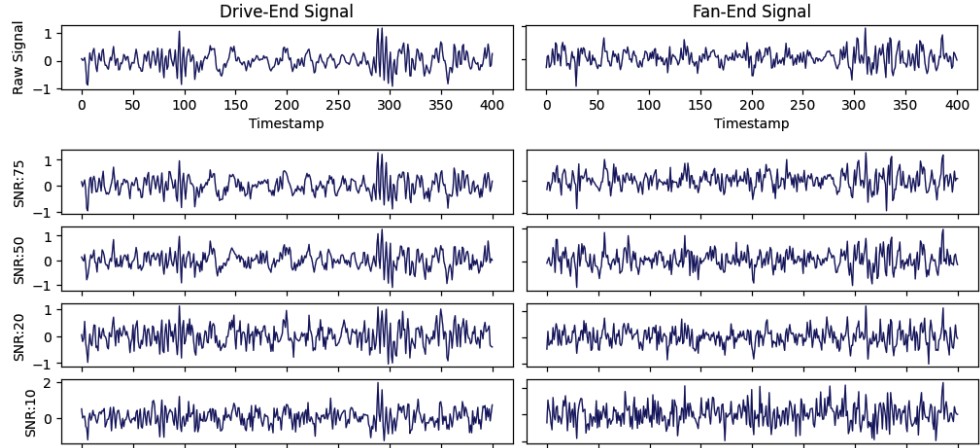

**Figure 7.** Some noisy signal samples generated from raw sensory data with different SNRs.

*5.2. GAN Model Selection*

As it is mentioned in the previous section, the proposition of Wasserstein loss function and adding the gradient penalty to its loss function help stabilize the generative algorithm. Figure 8 depicts how the proposed WGAN-GP reaches an equilibrium after 9000 epochs where it can generate realistic samples. Whereas, the other GAN generators make samples

which cannot devise their discriminators. As it can be clearly seen in Figure 8 their generator loss values go significantly higher than those of the discriminators. This comparison demonstrates why the implementation of WGAN-GP is preferred. Figure 9 shows some real samples of normal baseline, and fault conditions associated with the bearing ball, inner race and outer race with fault diameters of 7 mils and 21 mils. Figure 10, similarly, visualizes the synthetic samples generated by WGAN-GP after 10,000 epochs.

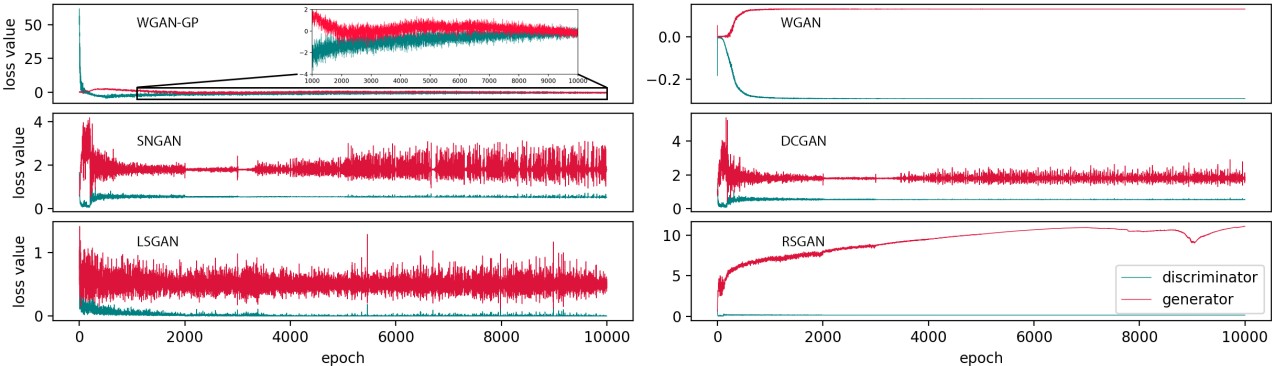

**Figure 8.** The generator and discriminator loss values for different GAN architectures.

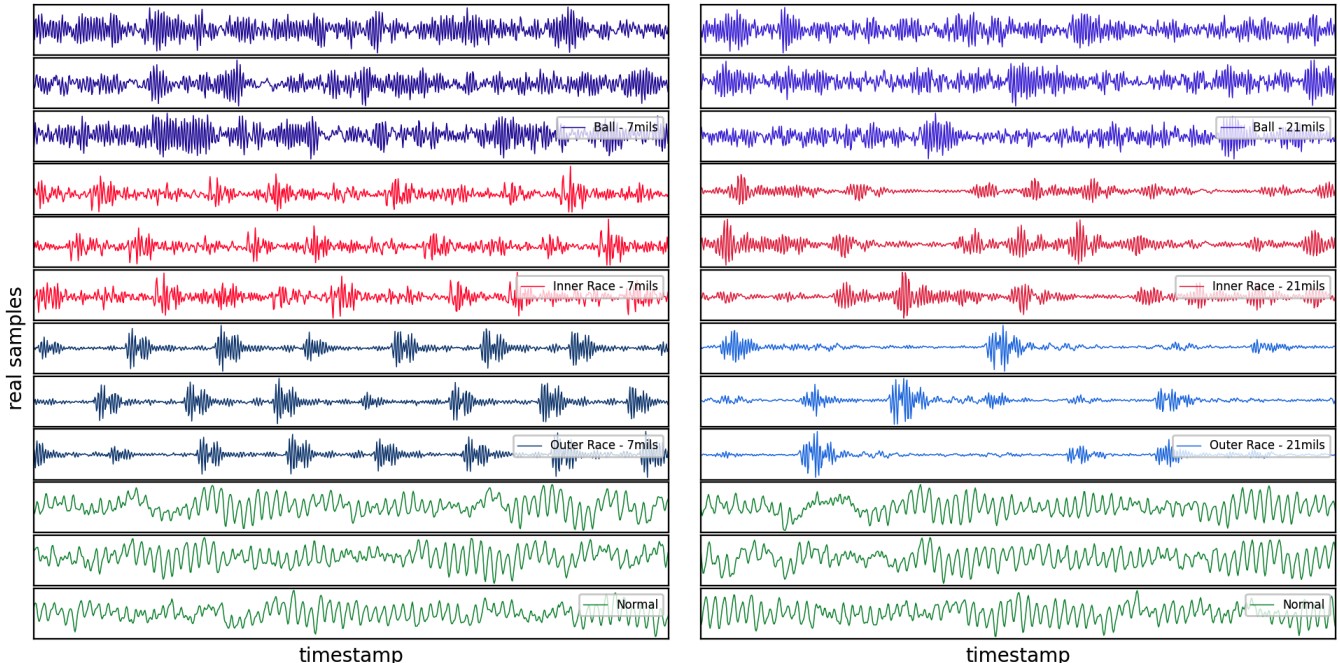

**Figure 9.** Some real samples associated with different running conditions.

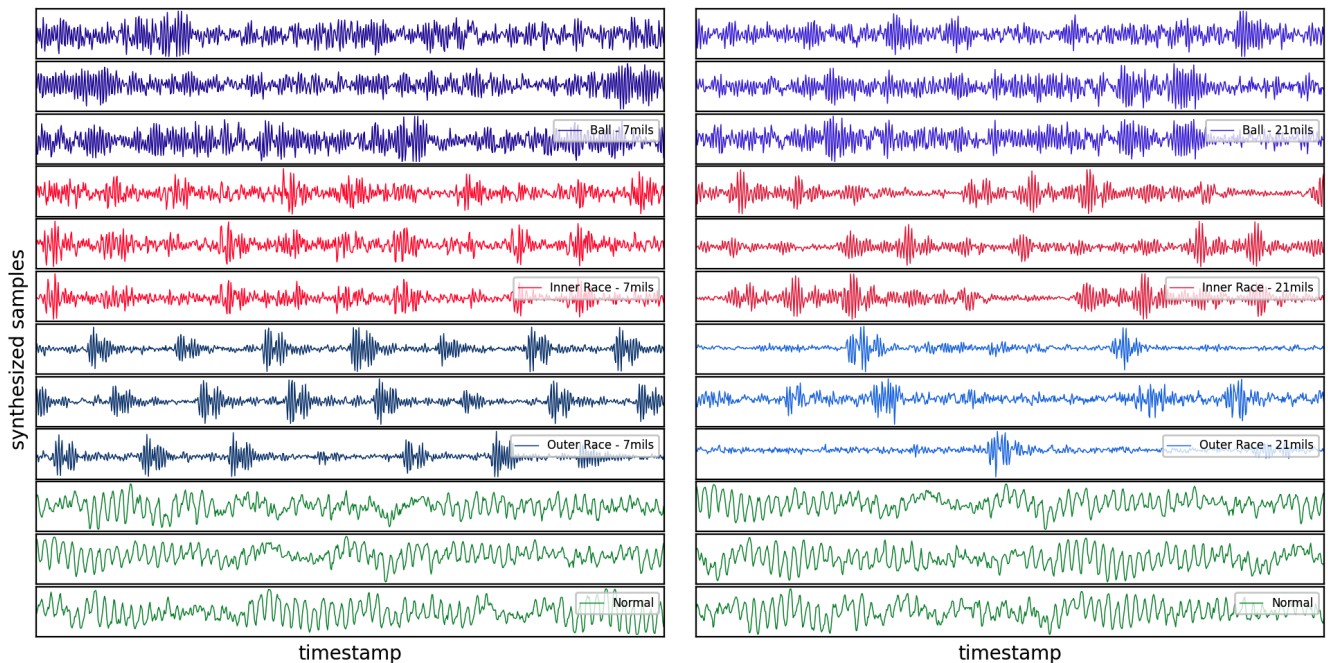

**Figure 10.** Some random synthesized samples associated with different running conditions made by WGAN-GP.

### 5.3. The Sensitivity Analysis

In this section the paper illustrates a sensitivity analysis on the performance of the proposed model by changing the $\alpha$ values and the SNRs. Specifically, we considered 25 points with respect to $\alpha = 2^k : k = -2, -1, 0, 1, 2$ and SNR $= (10, 20, 50, 75, 100)$, and run the model 10 times at each point to achieve a robust analysis. Figures 11 and 12 demonstrate the performance of GAN-CLSTM-ELM model with different metrics on these points.

As it can be seen in the figures, high levels of noise impact on the performance of the model, changing the $f_1$ score from 100% to 95.91%, and from 99.7% to 81.45% when $\alpha = 4$ and $\alpha = 0.25$, respectively. In this defined space the accuracy, AUC and recall values fall above 96.7%, 92.6% and 81.16%, respectively. The model shows a relatively high robustness to both noise and imbalance severities for SNRs greater than 20. At its best-case scenario, where $\alpha = 4$ and SNR $= 100$, it gains $f_1$ score of 100%; in its worst-case scenario, where $\alpha = 0.25$ and SNR $= 50$, it respectively gets 98.02% and 99.77% of $f_1$ score and accuracy. In the following, the paper conducts a comparison to figure out how these numbers are meaningful and whether the proposed model can better mitigate the adverse impacts of imbalanced and noisy conditions.

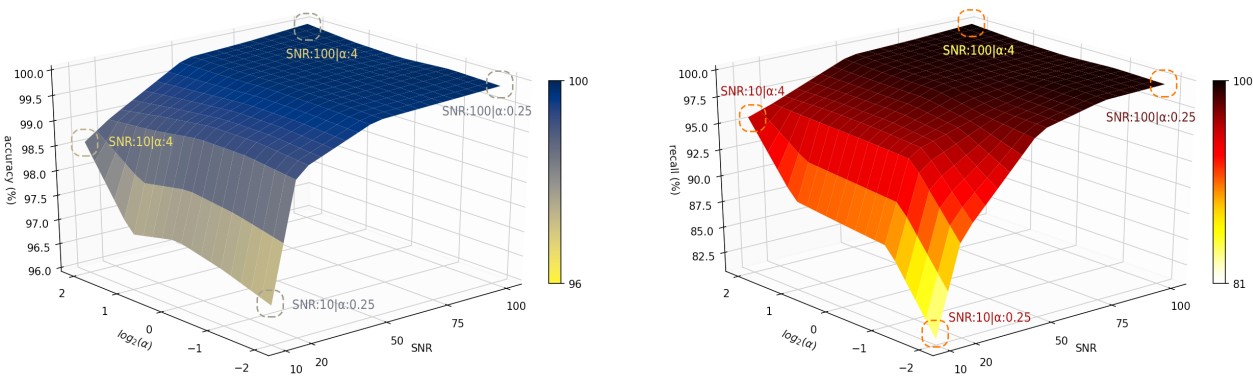

**Figure 11.** Accuracy (**left**) and Recall (**right**) performances of GAN-CLSTM-ELM in different SNR and $\alpha$ levels.

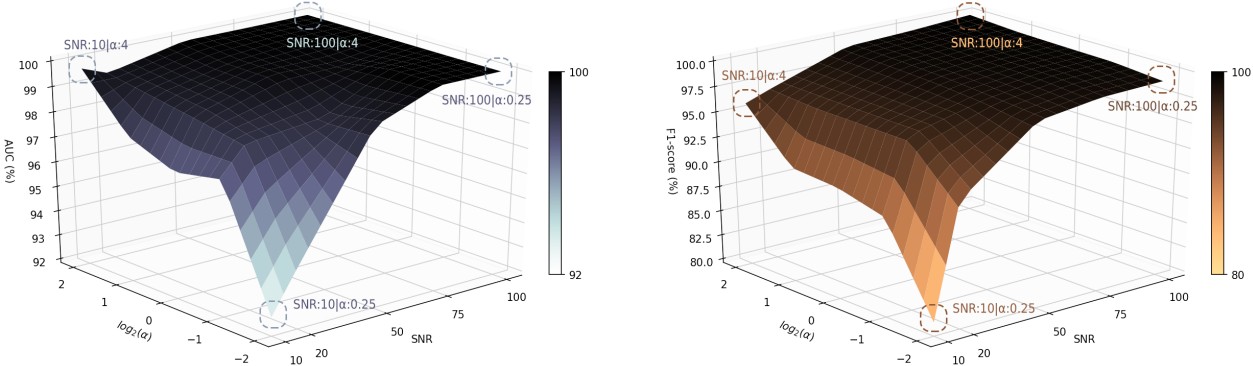

**Figure 12.** AUC (**left**) and $f_1$ score (**right**) performances of GAN-CLSTM-ELM in different SNR and $\alpha$ levels.

### 5.4. Model Performance Evaluation

In order to achieve meaningful comparisons, some novel FDD frameworks were employed to perform the diagnosis at different scenarios. CLSTM, df-CNN, sdAE, WELM, and CNN have shown promising performances in the literature, hence, they were selected for this purpose. Three traditional machine learning classifiers, SVM, ANN and Random Forest (RF), are also considered in this experimental comparison to draw insights from both machine learning and deep learning models. CLSTM-ELM was also added to the comparison panel to examine the necessity of Weighted ELM in the architecture of the proposed framework. A grid search was designed on the hyper-parameters of these models to achieve higher performances. Specifically, the learning rate, batch size and the architecture of fully connected layers were optimized for each algorithm. This paper uses two augmentation techniques: (i) "classic": where the samples are flipped, mirrored and different white noises are added to them. (ii) "GAN": with WGAN-GP, as discussed earlier in Section 4. all the frameworks are examined with both augmentation techniques. A brief description of the selected frameworks is provided in Table 3.

To avoid the weight initialization effect and randomness on the results, we ran each framework for ten independent times, using a five-fold cross validation technique on each imbalance and noise degree conditions. The data is stratified, such that each fold has the same class distribution. For the minority class, depending on the $\alpha$ value, there will be between 51 and 816 samples on which it can train. The classic augmentation technique is used to multiply this number by 8, (mirroring and flipping the samples and adding random white noise to them). In each scenario after training the WGAN-GP on each class, we set it to produce between 512 to 4096 samples for the minority class such that its sample number matches the other classes.

Figures 13 and 14 illustrate the corresponding normalized confusion matrices and the model performances with both classic and WGAN-GP augmentations, respectively. Comparing the different scenarios, it can plainly be concluded that GAN-CLSTM-ELM has a better ability to extenuate the negative effects of imbalance and noise conditions compared to the other frameworks. Regarding the highly imbalanced situation, its $f_1$ score has gently dropped by 0.32% in the first two scenarios (SNR:100, $\alpha$:$2^{-2}$ and SNR:100, $\alpha$:$2^2$) while the other frameworks have shown relatively substantial declines in their $f_1$ scores, ranging from 1.14% (GAN-CNN) to roughly 48% (df-CNN). In the second scenario, while the proposed model correctly identifies all the minority class samples, CLSTM-ELM and GAN-CNN were able to classify roughly 92% of them. This percentage for CLSTM, sdAE, WELM and CNN was between 80 and 85. The df-CNN showed a lackluster performance on the minority class as it could not correctly diagnose any of the corresponding samples. The figures also show that replacing the fully connected layers with W-ELM in the CLSTM-ELM model has slightly increased its robustness when $\alpha$ plummets from 4 to 0.25.

**Table 3.** The comparison panel.

| Framework | Preprossecing | Description | References |
|---|---|---|---|
| CLSTM | FFT + CWT + Statistical features | Its architecture comprises two CNN blocks (containing 1D-Convolutional layers, Batch Normalization, ReLU and Max Pooling), a LSTM block, a Logarithmic SoftMax, a concatenation which adds statistical features and three fully connected neural networks for the classification. | [46] |
| CLSTM-ELM | FFT + CWT + Statistical features | Its CNN and LSTM architecture are the same as in CLSTM; yet, the fully connected layers are substituted for W-ELM with 150 nodes. | N/A |
| df-CNN | raw signals | It is proposed to make an abstract 2-dimensional image out of raw signals. Its architecture comprises two CNN blocks (containing 2D-Convolutional layers, Batch Normalization, ReLU and Max Pooling), and three fully connected neural networks for the classification. df-CNN works directly on the raw vibration signals. | [48] |
| sdAE | raw signals | It is a multilayered architecture composed of four auto-associative neural network layers, which contain one input layer and three AEs. The input of this framework are raw signals. | [14] |
| CNN | FFT | The architecture consists of three CNN blocks (containing one 1D-Convolutional layer and a Pooling layer), two fully connected layers, and a SoftMax classification layer. It takes short-term Fourier transform (STFT) form of the signals as its input. | [16,31,33] |
| W-ELM | FFT + VMD + Statistical features | It takes a combination of FFT, VMD [49] and some statistical features. | [20] |
| SVM | Statistical features | SVM with polynomial kernel and degree of 2 is selected | N/A |
| ANN | Statistical features | 3 fully connected layers with a grid search to find optimal number of neurons per layer and the activation functions | N/A |
| RF | Statistical features | A grid search is designed to find the optimal number of estimators, and criteria (between 'gini' and 'entropy') parameters | N/A |

In the presence of heavy noises, there are sudden falls in the performances of all the algorithms. Comparing the first and the third scenarios (SNR = 100, $\alpha = 2^2$ and SNR = 10, $\alpha = 2^2$), all the CLSTM-based methods alongside GAN-CNN had the least decrease in the $f_1$ score (roughly 5%); thus, they were the most robust algorithms in noisy conditions. Comparing CLSTM-ELM and CLSTM with CNN, in both figures, we can infer that the presence of LSTM and CWT, has made the model perform better against the noise. Moreover, CNN achieved comparatively poorer results when $\alpha$ dips below 1. Its combination with a WGAN-GP, however, mitigated this loss and GAN-CNN achieved a satisfactory result. With the presence of heavy noises, GAN-WELM classification quality drastically plunged and, despite its comparatively satisfactory performance in the first two scenarios, the noise made it unable to diagnose the minority class in highly imbalanced situations.

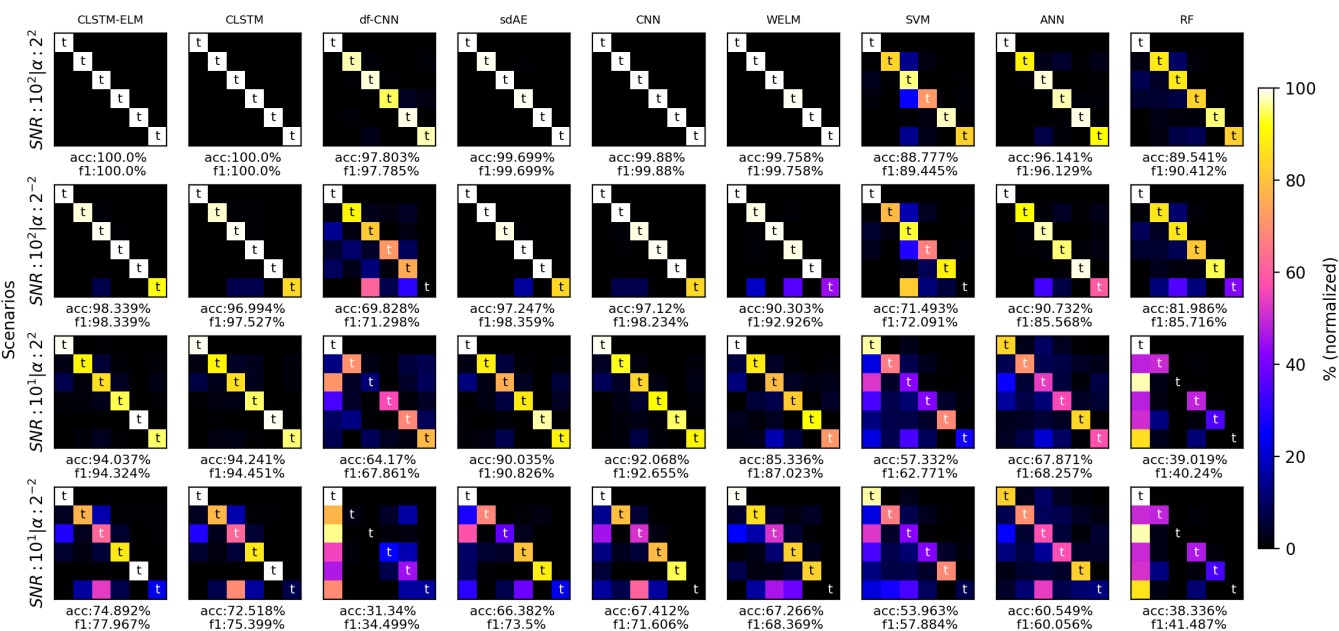

**Figure 13.** Confusion matrices and $f_1$-scores of the comparison panel with classic augmentation in different scenarios (t represents true labeled samples).

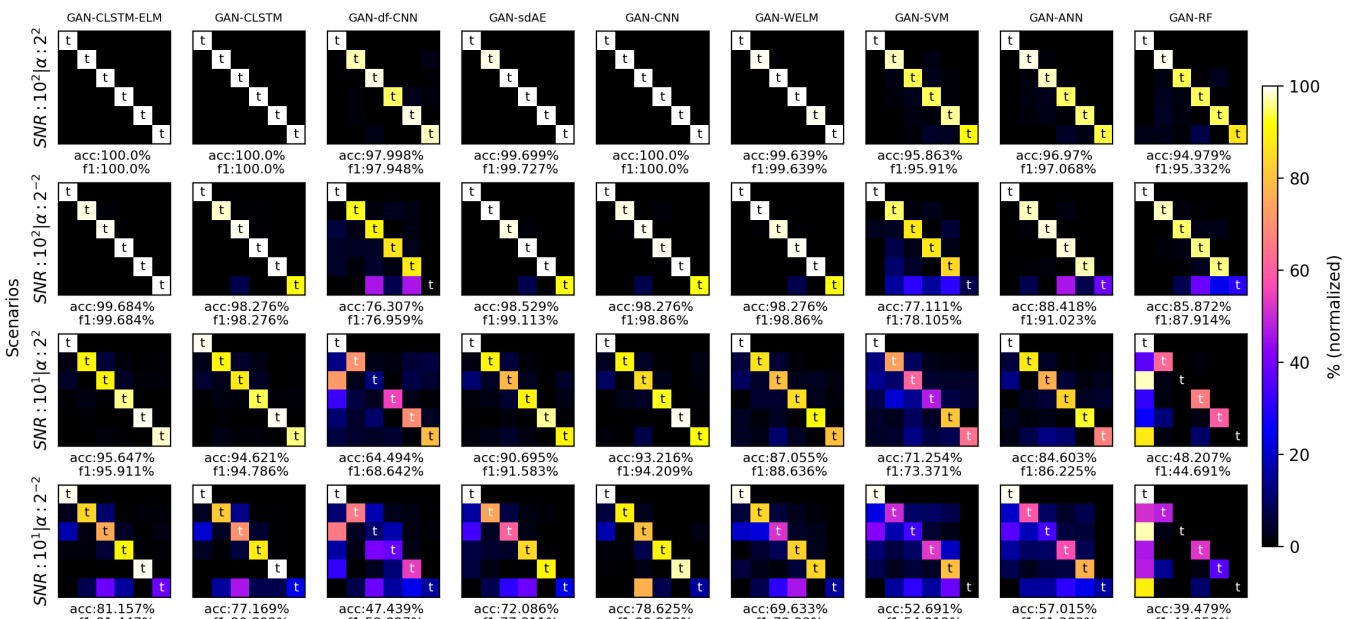

**Figure 14.** Confusion matrices and $f_1$-scores of the comparison panel with WGAN-GP augmentation in different scenarios (t represents true labeled samples).

By comparing the confusion matrices of WELM and CLSTM-ELM, it can be concluded that CLSTM architecture alongside WELM model improves its performance against the noise. It is worth noting that, adding WGAN-GP to the deep learning-based models, made them exhibit superiority over their root algorithms. This proves that WGAN-GP can effectively enhance the quality of the classifier not only in imbalanced situations but also in noisy environments. On the other hand, shallow learning techniques had comparably higher misclassification rates when it comes to either noisy or imbalanced conditions. GAN-based augmentation significantly improved RF and ANN accuracy scores, except for the SVM, as it was unable to diagnose the minority class in highly imbalanced situations.

Table 4 shows each deep learning algorithm training time per step and the learning hyper-parameters. As it is discussed in [46], CLSTM has a relatively slow training phase. From the table, it can be seen that substituting the W-ELM for fully-connected layers has made it slightly faster to train and converge. Among the comparison panel, df-CNN followed by WELM and CNN were the quickest classifiers. From Table 5, it can be inferred that the presence of noise makes the computations harder for the SVM and the RF to classify the samples. Their average training times were, therefore, drastically dependent on the scenarios. While, the deep learning-based classifiers had steady runtimes in different situations.

**Table 4.** Runtime comparison of the deep learning classifiers.

| Algorithm | Runtime/Step (ms) | #Epochs to Converge | Batch Size | Learning Rate |
|---|---|---|---|---|
| CLSTM-ELM | $104.7 \pm 8.6$ | 10 | 64 | $10^{-3}$ |
| CLSTM | $116.4 \pm 10.5$ | 12 | 64 | $10^{-3}$ |
| df-CNN | $12.1 \pm 1.8$ | 8 | 64 | $10^{-3}$ |
| sdAE | $43.3 \pm 4.1$ | 6 | 64 | $10^{-4}$ |
| CNN | $21.8 \pm 1.7$ | 7 | 64 | $10^{-4}$ |
| WELM | $17.2 \pm 1.4$ | 7 | 32 | $10^{-3}$ |

**Table 5.** Training times of the shallow learning-based classifiers.

| Algorithm | Training Time (min) | | | |
|---|---|---|---|---|
| | $SNR = 10^2 \| \alpha = 2^2$ | $SNR = 10^2 \| \alpha = 2^{-2}$ | $SNR = 10^1 \| \alpha = 2^2$ | $SNR = 10^1 \| \alpha = 2^{-2}$ |
| SVM | $8.73 \pm 0.45$ | $8.90 \pm 0.37$ | $18.67 \pm 1.32$ | $19.40 \pm 1.49$ |
| ANN | $1.97 \pm 0.22$ | $1.72 \pm 0.31$ | $2.06 \pm 0.38$ | $2.17 \pm 0.42$ |
| RF | $2.13 \pm 0.40$ | $1.55 \pm 0.36$ | $3.06 \pm 0.33$ | $3.21 \pm 0.26$ |

## 6. Discussion and Conclusions

In many real applications of fault detection and diagnosis data tend to be imbalanced and noisy, meaning that the number of samples for some fault classes is much fewer than the normal data samples and there are errors in recording the actual measurement by the sensors. These two conditions make many traditional FDD frameworks perform poorly in real-world industrial environments.

In this paper a novel framework called GAN-CLSTM-ELM is proposed, which enhances the performance of rotating machinery FDD systems coping with highly-imbalanced and noisy datasets. In this framework, WGAN-GP is first applied to augment the minority class and enhance the training set. A hybrid classifier is then developed, containing Convolutional LSTM and Weighted ELM, which learns more efficiently from vibration signals. The framework also benefits from both wavelet and Fourier transform techniques in its feature engineering step, revealing more hidden information of the fault signatures to make the classifier perform more accurately. The effectiveness of the proposed framework is verified by using four dataset settings with different imbalance severities and SNRs. After conducting the comparisons with state-of-the-art FDD algorithms, it is demonstrated that the GAN-CLSTM-ELM framework can reduce the misclassification rate and outperform the other methods, more significantly when the imbalance degree is higher. The efficiency of the WGAN-GP is also proved by comparing the results of the proposed model and CLSTM-ELM as well as all the other diagnosis models. The experimental results make it discernible that using a generative algorithm helps the classification model alleviate the adverse impacts of low SNRs. Therefore, it stresses the necessity of employing such hybrid frameworks for practitioners working on noisy and industrial applications. The paper also justifies the implementation of W-ELM in the architecture of CLSTM, since the adjusted model shows sturdy classification when $\alpha$ decreases either in noisy or noiseless scenarios. A sensitivity analysis is designed with 25 dataset settings built on a range of $\alpha$ and SNR values, to obtain insights of how these two factors impact on the model's classification ability.

Extracting the FFT and CWT spectra needs some knowledge of signal processing and is still more convenient than extracting other hand-crafted features proposed in the literature. Another advantage of the proposed framework is that it gains comparatively high performances under noisy conditions while it requires no complex denoising preprocessing being handled by employees with expert knowledge of signal processing. These characteristics make GAN-CLSTM-ELM an attractive option for industrial practitioners who are in need of a relatively easy-to-use software without undergoing any complicated pre-processing task.

Future work will include more experiments on the behavior of different generative algorithms and the development of a more powerful architecture to create high-quality signals with fewer samples. We will also attempt to explore the feasibility of implementing and testing the proposed framework on other applications.

**Author Contributions:** Conceptualization, M.J. and C.O.; methodology, M.J.; software, M.J.; validation, M.J., A.K. and C.O.; formal analysis, M.J.; investigation, M.J.; data curation, M.J.; writing—original draft preparation, M.J. and A.K.; writing—review and editing, C.O.; visualization, M.J.; supervision, C.O. and C.V. All authors have read and agreed to the published version of the manuscript.

**Funding:** This research received no external funding.

**Institutional Review Board Statement:** Not applicable.

**Informed Consent Statement:** Not applicable.

**Data Availability Statement:** https://engineering.case.edu/bearingdatacenter (accessed on 22 March 2022).

**Conflicts of Interest:** The authors declare no conflict of interest.

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
