# Peer review of "Fault Detection and Diagnosis with Imbalanced and Noisy Data: A Hybrid Framework for Rotating Machinery"

_machines, doi:10.3390/machines10040237_

Round 1
Reviewer 1 Report
The paper “Fault Detection and Diagnosis with Imbalanced and Noisy Data: A Hybrid Framework for Rotating Machinery” proposes an approach for faults detection and classification in rotatory machinery. The approach includes a stage for vibration data augmentation based on the Wasserstein Generative Adversarial with Gradient Penalty Networks (WGAN-GP). Additionally, authors propose a classification stage that combines convolutional neural networks and Long Short-term Memory (LSTM) whose output is fed to a Weighted Extreme Learning Machine (WELM) for faults classification. The classification is fed with features extracted using Fast Fourier transform and the continuous wavelet transform. The approach is validated using the CWRU bearing dataset.
Authors discuss in the introduction section the problem of faults detection in rotating machinery using relevant references. They also discuss the main contributions of their research. The paper includes in the section 2 a discussion concerning the state of the art concerning the application of machine learning models to de problem of faults rotation in rotating machinery as well as the applications of data augmentation. The section is mainly centered in the analysis of Deep Learning applications.
The paper in general is clearly written, and the approach includes a validation and comparison with several methods. There are however several issues that should be corrected before publication. The list is presented below:
1) It is not clear how the CWT and FFT information is merged. In Figure 3 authors claim using a FFT processor and a CWT processor. However, they do not provide any detail concerning this stage. Authors should explain this stage in the paper. The parameters used in this stage should be presented, for instance, the type of wavelet. Authors should include examples of the CWT (and FFT) for each type of faults. Authors should also present the size of the matrix representing this information. Similarly, it is no clear if the FFT processor obtain STFT? What is the type of window? What is the size of the window? What is the size of the matrix? If both types of information are 2D, several schemes of concatenation are possible. Authors should explain how the concatenation of both types of information is performed.
2) In Figure 3 authors should also provide information concerning the size of each network, CNN and LSTM.
3) Authors should provide details concerning the number of iterations, and learning factors for each model. Authors should compare the convergence ability of the proposed method with respect to other methods.
4) Authors claim to use k-fold cross-validation, however they do not provide details concerning the number of folds used. Authors should better describe the dataset used for validation in terms of training set and test set. After augmentation, how many examples for each class are included. How long is each signal?
5) Authors should include the time required for training each of the models compared.
6) Authors should explain in the paper why two different types of augmentation methods are used in combination with the classification method (in Table 2). It should be better to compare all the classification methods with the same type of augmentation method. It means constructing one figure similar to figure 11 using classic method and then other figure considering the proposed method WGAN-GP.
Reviewer 2 Report
Under industrial conditions, the data of fault detection and diagnosis tend to be imbalanced, and the collected samples will have a lot of noise. This paper proposes a hybrid framework called GAN-CLSTM-ELM, which improves the performance of rotating machinery fault detection and diagnosis system in dealing with highly unbalanced and noisy datasets. The experimental results on different datasets illustrated the effectiveness of the proposed framework.
- It is recommended to place Figure 8 on page 14 below Figure 7 for easy reading;
- Whether "SNR: 100|α: 0.25" is marked incorrectly in the first picture of Figure 9 and Figure 10 on page 14;
- Line 435 on page 16 mentions “comparing the first and the third scenarios... they were the most robust algorithms in noisy conditions.”, however, it can be seen from Figure 11 that the F1 score of CLSTM algorithm also decreased by about 5%.
Round 2
Reviewer 1 Report
The paper “Fault Detection and Diagnosis with Imbalanced and Noisy Data: A Hybrid Framework for Rotating Machinery” proposes an approach for faults detection and classification in rotatory machinery. The approach includes a stage for vibration data augmentation based on the Wasserstein Generative Adversarial with Gradient Penalty Networks (WGAN-GP). Additionally, authors propose a classification stage that combines convolutional neural networks and Long Short-term Memory (LSTM) whose output is fed to a Weighted Extreme Learning Machine (WELM) for faults classification. The classification is fed with features extracted using Fast Fourier transform and the continuous wavelet transform. The approach is validated using the CWRU bearing dataset.
Authors discuss in the introduction section the problem of faults detection in rotating machinery using relevant references. They also discuss the main contributions of their research. The paper includes in the section 2 a discussion concerning the state of the art concerning the application of machine learning models to de problem of faults rotation in rotating machinery as well as the applications of data augmentation. The section is mainly centered in the analysis of Deep Learning applications.
The paper in general is clearly written, and the approach includes a validation and comparison with several methods. Authors have incorporated the corrections that were indicated in the previous round and, in my opinion, the paper has been improved and it is ready for publication.